# Design of Antibody-Functionalized Polymeric Membranes for the Immunoisolation of Pancreatic Islets

**Anna Cavallo [1], Ugo Masullo [1], Alessandra Quarta [2,*], Alessandro Sannino [1], Amilcare Barca [3], Tiziano Verri [3], Marta Madaghiele [1] and Laura Blasi [2,4,*]**

[1]  Department of Engineering for Innovation, University of Salento, Via Monteroni, 73100 Lecce, Italy; anna.cavallo@unisalento.it (A.C.); ugo.masullo@hotmail.com (U.M.); alessandro.sannino@unisalento.it (A.S.); marta.madaghiele@unisalento.it (M.M.)
[2]  CNR Nanotec, Institute of Nanotechnology, via Monteroni, 73100 Lecce, Italy
[3]  Department of Biological and Environmental Sciences and Tecnologies, University of Salento, 73100 Lecce, Italy; amilcare.barca@unisalento.it (A.B.); tiziano.verri@unisalento.it (T.V.)
[4]  CNR-IMM, Institute for Microelectronics and Microsystems, Via Monteroni, 73100 Lecce, Italy
*  Correspondence: alessandra.quarta@nanotec.cnr.it (A.Q.); laura.blasi@le.imm.cnr.it (L.B.)



**Featured Application: Reducing the risk of rejection in patients transplanted with pancreatic islets through their immuno-encapsulation with antibody-conjugated polymeric membranes.**

**Abstract:** An immunoencapsulation strategy for pancreatic islets aimed to reduce the risk of rejection in transplanted patients due to the immune response of the host organism is proposed. In this sense, a polyethylene glycol (PEG) hydrogel functionalized with an immunosuppressive antibody (Ab), such as Cytotoxic T-lymphocyte antigen-4 Ig (CTLA4-Ig), would act as both passive and active barrier to the host immune response. To demonstrate the feasibility of this approach, a photopolymerizable-PEG was conjugated to the selected antibody and the PEG-Ab complex was used to coat the islets. Moreover, to preserve the antigen-recognition site of the antibody during the conjugation process, a controlled immobilization method was setup through the attachment of the His-tagged antigen to a solid support. In detail, a gold-coated silicon wafer functionalized with 11-Mercaptoundecanoic acid was used as a substrate for further modification, leading to a nickel(II)-terminated ligand surface. Then, the immobilized antigen was recognized by the corresponding antibody that was conjugated to the PEG. The antibody-PEG complex was detached from the support prior to be photopolymerized around the islets. First, this immobilization method has been demonstrated for the green fluorescent protein (GFP)–anti-green fluorescent protein (Anti-GFP) antigen-antibody pair, as proof of principle. Then, the approach was extended to the immunorelevant B7-1 CTLA-4-Ig antigen-antibody pair, followed by the binding of Acryl-PEG to the immobilized constant region of the antibody. In both cases, after using an elution protocol, only a partial recovery of the antibody-PEG complex was obtained. Nevertheless, the viability and the functional activity of the encapsulated islets, as determined by the glucose-stimulated insulin secretion (GSIS) assay, showed the good compatibility of this approach.

**Keywords:** polyethylene glycol; immunoisolation; pancreatic islets

## 1. Introduction

Type 1 diabetes is a worldwide debilitating autoimmune disease, in which β cells within pancreatic islets are selectively destroyed by autoimmune responses against β cells [1]. Among available therapies, islet transplantation can treat the most severe cases of type 1 diabetes, although the shortage of islet

donors and the requirement of a chronic immunosuppression to prevent rejection and recurrence of autoimmunity limit its applicability [2]. In this context, tissue engineering and regenerative medicine offer new strategies to circumvent these problems by properly modifying the graft before transplantation, in order to achieve immunoexclusion and, thus, limit the need for immunosuppressive treatments [3–5]. Modifications are focused on improving viability, functionality, and protection of the graft through the creation of a suitable microenvironment for the implant, thus, favoring the engraftment of the islets [6,7]. Immunoencapsulation is usually achieved by means of approaches that permit exchange of oxygen, nutrients, insulin, and waste, and, at the same time, protect those islets from the immune response [3,8].

The employment of thin polymeric layers reduces the size of the coating and the invasiveness of the process with respect to other approaches, such as the development of macrodevices and the microencapsulation of islet cells [3,9,10]. Indeed, the main drawbacks of these approaches are related to the size of the coating (500 to >1000 μm) that limits oxygen diffusion to the core of the capsule, thus favoring the survival of the islets that are closed to the surface [11,12].

On the other hand, conformal coating facilitates oxygen and nutrient diffusion to support islet survival and function after implantation, and enables a physiologic release of insulin into the bloodstream compared to microcapsule [13,14]. Among conformal coating studies, the use of polyethylene glycol (PEG) hydrogels obtained by photopolymerization for islet immunoisolation showed great promise in rodents [15–17]. Indeed, PEG hydrogels display soft tissue-like properties, non-immunogenicity, the intrinsic resistance to protein adsorption, and the suitability for being covalently linked to multiple functional moieties [18,19].

However, one of the fundamental deficiencies with current encapsulation technology is the evidence that passive physical barriers, while preventing T-cell infiltration and cell contact-mediated destruction of grafted islets, cannot protect islets from exposure to local cytokines and other small, diffusible cytotoxic molecules produced by activated immune cells [20,21].

To this aim, PEG was exploited as a substrate to conjugate immunomodulatory antibodies or bioactive molecules, potentially able to enhance the immunoprotection activity and drive targeted immunosuppression. Indeed, we propose a PEG-antibody functionalization protocol that preserves the antigen-recognition site, namely a site-directed functionalization. Non-specific physical adsorption or chemical crosslinking results in the immobilization of the biomolecules in a random orientation. Site-specific covalent attachment, on the other hand, leads to molecules being arranged in a definite and oriented way [22].

The proposed functionalization strategy exploits the selective, oriented, and reversible immobilization of a recombinant protein (that is the antigen) bearing a polyhistidine-tag (His-tag) to a transition metal terminated nitrilotriacetate group derivatized support [23,24]. Then, the site-oriented immobilization of the antibody to the antigen enables the functionalization of the constant region of the antibody with PEG, followed by the recovery of the antibody-PEG complex upon breaking of the antigen-antibody bond. Therefore, the recovered PEG-functionalized antibody displays the binding site, being available to exert its bio-protection activity upon coating of the pancreatic islets.

Initially, as a proof of concept, the site oriented functionalization method was investigated with a fluorescent antigen-antibody (Ag-Ab) pair, i.e., green fluorescent protein (GFP)–anti-green fluorescent protein antibody (Anti-GFP): the fluorescence emission of the antigen facilitated the identification of the corresponding Ab bound to the PEG hydrogel. Finally, an immune relevant Ag-Ab pair, namely B7-1 His-tagged (B7-1) and Cytotoxic T-lymphocyte antigen-4 Ig (CTLA-4 Ig), was used to demonstrate the feasibility of this approach [25]. The immunosuppressive behavior of CTLA-4 Ig has been indeed discovered and the use of this antibody as therapeutic drug in some immune-based diseases already proposed [26–28].

## 2. Materials and Methods

### 2.1. Chemicals

The 11-Mercaptoundecanoic acid (MUA), *N*-(3-Dimethylaminopropyl)-*N'*-ethylcarbodiimide hydrochloride (EDC), *N*-Hydroxysulfosuccinimide (NHS), Nickel(II) chloride (NiCl2), *N*α,*N*α-Bis(carboxymethyl)-L-lysine hydrate (NTA-lysine), and Acryl-PEG-NHS (Mw 3000 Da) were purchased from Sigma-Aldrich. Green fluorescent protein (GFP) with *N*-terminal His-Tag (~28kDa) and chicken anti-green fluorescent protein antibody (Anti-GFP) were purchased from Millipore. Goat Anti-Chicken IgY (H+L) HRP-conjugated, and Alexa Fluor® 488 goat Anti-Chicken and Alexa Fluor® 647 goat Anti-Chicken were purchased from Thermo Scientific. Recombinant Human B7-1 His-tagged (B7-1) was purchased from R&D Systems. CTLA-4-Ig was purchased from Abcam.

### 2.2. Site-Directed Functionalization of the Antibody and Characterization Procedures

*Preparation of the substrate for the immobilization of the protein antigen*. The substrates were prepared by depositing a film of gold with a thickness of 250 nm (± 50 nm) on the top of an adhesive chromium layer (2.5 nm) deposited on a $SiO_2$/Si (100 nm) wafer; afterwards, squared chips (area of about 1 $cm^2$) were cleaved using a diamond-tipped scriber. The samples were rinsed with acetone, isopropyl alcohol, and ethanol (at least three times for each solvent) and blown dry with pure nitrogen. The gold surface was modified using a three-step procedure (Figure 1), before the immobilization of His-tagged protein.

Gold-coated chips were immersed in a 10 mM solution of 11-mercaptoundecanoic acid in ethanol (EtOH) overnight (ON) at room temperature (RT). Then, the substrates were rinsed with EtOH in order to remove MUA molecules not bound to the surface and dried with nitrogen (Figure 1a). The activation and amidation of the carboxylate groups on Au-MUA substrates were carried out in a double step process: firstly, the samples were exposed to 100 mM *N*-(3-Dimethylaminopropyl)-*N'*-ethylcarbodiimide hydrochloride (EDC) and 20 mM *N*-Hydroxysulfosuccinimide (NHS) aqueous solution for 1.5 h at RT. Then, they were immersed in 40 mM *N*α,*N*α-Bis(carboxymethyl)-L-lysine hydrate (NTA-lysine) aqueous solution (2 h at RT). Finally, the substrates were thoroughly rinsed with 150 mM NaCl and 10 mM HEPES aqueous solution (Figure 1b). The amino-nitrilotriacetic-Ni(II) complex (NTA-$Ni^{2+}$) was formed by reaction of Au-MUA-NTA-lysine with nickel(II) chloride (1 mM) in 150 mM NaCl and 10 mM HEPES buffer pH 7.4 (2 h at RT) (Figure 1c).

*Atomic Force Microscope (AFM) characterization*. Non-Contact mode Atomic Force Microscope (NC-AFM, XE-100 PSIA, Suwon, Korea) analysis was conducted to monitor the formation of MUA film onto gold surfaces. The Root Mean Square Roughness of AFM images is calculated on the basis of three independent measurements.

*Contact angle measurements*. The water contact angle (WCA) of the self-assembled monolayer (SAM) on gold surface was measured with a KSVCAM200, Kruss, Hamburg, Germany, contact angle goniometer, using distilled water droplets of 2 μL. The average contact angle values with the corresponding standard deviations are reported for each sample based on five measurements. Measurements were carried out on the gold-coated substrates before MUA incubation, after 12 h incubation (MUA1), after 60 h incubation (MUA2), and after the double step of MUA2 and NTA-lysine incubation.

*Specific Immobilization of Histidine-Tagged antigen on Au-MUA-NTA-lysine-Ni(II), anchoring of target antibody and functionalization with Acryl-PEG-NHS*. To promote the attachment of the histidine tags bearing proteins to the transition metal complex, the modified substrates were incubated with either GFP or B7-1 (400 nM in 150 mM NaCl and 10 mM HEPES buffer) overnight (ON) at 4 °C and then rinsed abundantly with phosphate buffer saline (PBS) 1X.

Successively, the samples were exposed to a solution of the corresponding antibody (either anti-GFP or CTLA-4-Ig, 200 nM in PBS ON at 4 °C). Once the Ag-Ab interaction occurred, the antibody exhibited the constant fraction (Fc) on the top of the self-assembled monolayer (SAM). Activation of the carboxyl groups in Fc fragment and the subsequently bonding to Acryl-PEG-NHS were performed by

adding Acryl-PEG-NHS (2 μM) and EDC (20 μM) in PBS solution 4 h at 4 °C. Finally, the samples were abundantly rinsed with PBS in order to remove physically adsorbed molecules, and stored at 4 °C.

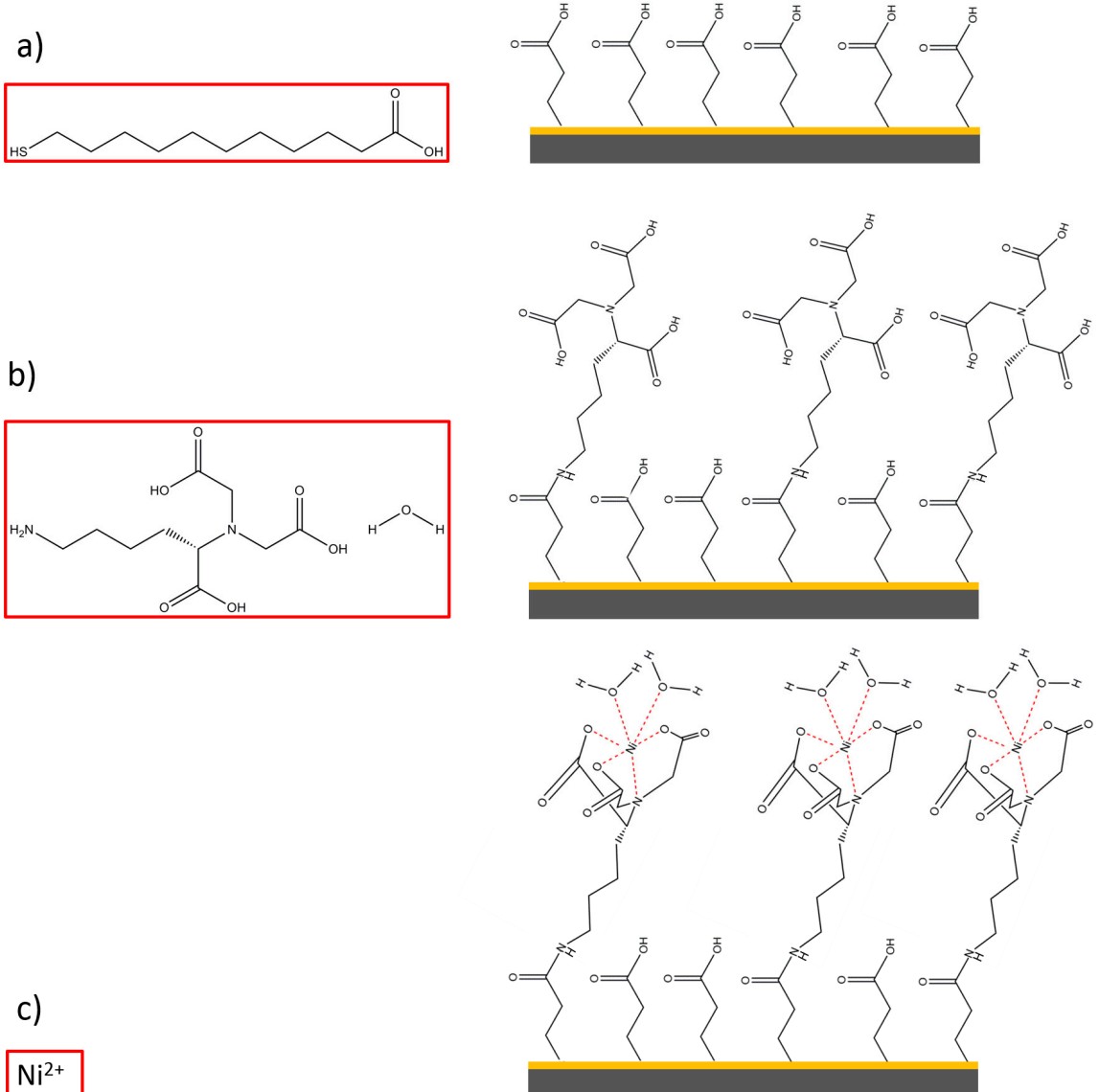

**Figure 1.** Chemical modification of the gold-coated silicon surface: (**a**) derivatization with MUA, (**b**) NTA-lysine binding (**c**) formation of the NTA-lysine-Ni²⁺ complex.

*Detachment and quantification of the protein antigen.* To quantify the protein attached to the SAM, the samples were exposed to the eluent buffer (150 mM NaCl, 10 mM HEPES, 500 mM Imidazole) for 1 h at 4 °C. The amount of the protein detached from the substrate and eluted was determined comparing the GFP fluorescent emission of eluted solution with the feeding solution (400 nM) as well as the recovered solution (that corresponds to the unbound protein fraction, recovered during the washing steps). Photoluminescence spectra were recorded using a Cary Eclipse spectrofluorometer. Then, the relative peak intensities (at 508 nm) of the eluted and recovered samples were compared to that of the feeding GFP solution and plotted in a graph.

*Western Blot analysis.* An alternative method to quantify the attached antigen was based on Western Blot assays. The proteins were run on 10% Tris-Glycine SDS/PAGE (Bio-Rad) under denaturing conditions and transferred to a PVDF membrane. After blocking with 5% (*w/v*) nonfat dry milk in 1X PBS/0.1% (*v/v*) Tween 20, the membrane was incubated over night at 4 °C with an antibody against

GFP (1:2500 in 5% (*w/v*) nonfat dry milk in PBS/0.1% Tween 20 (*v/v*)). The membrane was then washed, incubated with anti-chicken peroxidase-conjugated secondary antibody (1:1000 in 1% (*w/v*) nonfat dry milk in PBS/0.1% (*v/v*) Tween 20) at room temperature for 1 h, and developed by ECL plus (Bio-Rad).

*Detachment and quantification of the Acryl-PEG-Antibody complex.* To elute the Acryl-PEG-Antibody (either Anti-GFP or CTLA-4-Ig) complex from the functionalized wafer, it was incubated with 0.1 M glycine-HCl, pH 2.8, at 4 °C for 10 min. This buffer dissociates most protein-protein and Ag-Ab binding complexes without permanently affecting protein structure [29]. Afterwards, the collected solution was concentrated by ultrafiltration with a 50 kDa centrifugal filter device (Amicon, Millipore). The Acryl-PEG-antibody complex was then quantified using Western Blot.

*Acryl-PEG-Anti-GFP photo-crosslinking on PEG hydrogel using a selective mask.* The functionality of acrylic groups on Acryl-PEG-Anti-GFP complex was assessed by photo-crosslinking, a hydrogel dish [30]. Briefly, a solution of Acryl-PEG-Anti-GFP 20 nM and Darocur® 1173 (0.3% *w/v*) was irradiated to conjugate the antibody to the hydrogel. A TEM grid was used as UV photomask to direct the radiation in limited and defined areas, and consequently to have a selective photopolymerization (Figure 2). Irradiation with a 365 nm UV light at 2 mW cm$^{-2}$ for 3 min was performed. During UV irradiation, the grid was in contact with the hydrogel to limit UV light diffraction due to air gap [31]. The presence of Acryl-PEG-Anti-GFP on the hydrogel was then visualized through a secondary FITC-conjugated antibody (Alexa Fluor® 488 anti-chicken, Thermo Fisher Scientific), incubated for 1 h at RT (1:250).

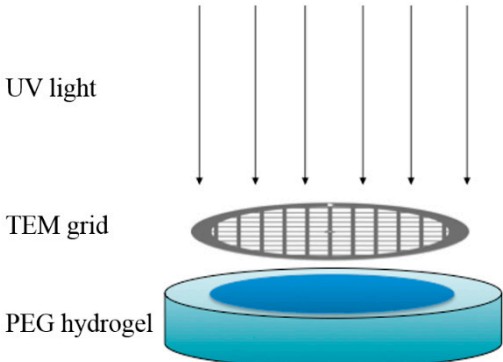

**Figure 2.** Scheme showing the selective photopolymerization of Acryl-PEG-Anti-GFP on a poly(ethylene) glycol diacrylate (PEGDA) hydrogel using a TEM grid as a mask.

### 2.3. Conformal Encapsulation of Pancreatic Islets in a PEG-Based Hydrogel

The procedure for the isolation of the pancreatic islets is reported in the Supporting Materials. A poly(ethylene) glycol diacrylate (PEGDA) solution was prepared by dissolving the PEGDA powder (Mw 6000 Da, Sigma-Aldrich) in sterile PBS Islets encapsulation as type 1 diabetes treatment 107 1X at a 10% *w/v* concentration with respect to the PBS 1X. A photoinitiator solution was prepared by dissolving Darocur 1173 in PBS 1X at a 3% w/w concentration with respect to the PEGDA. One day after isolation, islets were collected from the culture dish and washed once in PBS 1X. Then, islets were resuspended in 500 μL of the photoinitiator solution and incubated for 5 min at room temperature (RT) temperature in the dark, to allow the photoinitiator to adsorb onto their surfaces. After the incubation, the islets were immediately transferred to 150 μL of the PEGDA solution on an opened Petri dish and irradiated for 3 min under UV light (Jelolamp wood, Jelosil, 365 nm at 2 mW cm$^{-2}$). The UV irradiation allows the radical reaction to bond the acrylates together and propagate the crosslinking, producing an interfacially crosslinked hydrogel conformally structured around each islet.

### 2.4. Viability and Functional Assay of Pancreatic Islets

Viability of the encapsulated islets was assayed at 24 h post functionalization with Acryl-PEG-Anti-GFP, and was assessed using the fluorescein diacetate (FDA) *versus* propidium iodide (PI) staining

method. In detail, the islets were collected from the culture dish, washed and resuspended in PBS 1X, then PI and FDA solution were added individually to PBS 1X at final concentration of 20 µg/mL and 5 µg/mL, respectively, and incubated for about 30 min. Finally, confocal microscopy imaging (TCS-SP5, Leica, Mannheim, Germany) of the encapsulated and control islets was carried out to show the viability.

Islet functionality was determined by quantifying insulin release in response to glucose stimulation of the coated in comparison to uncoated islets through the glucose stimulated insulin secretion (GSIS). The assay was performed at day 1 after coating. About 10 islets for each sample were placed onto each Millicell cell culture insert (mesh 12 µm) (Millipore) and pre-incubated in a "low-glucose" (1.6 mM) Krebs–Ringer bicarbonate buffer (KRBB; 0.5% BSA, 2.5 mM $CaCl_2$, 4.7 mM KCl, 1.2 mM $KH_2PO_4$, 118.9 mM NaCl, 25.2 mM $NaHCO_3$, 2.5 mM $MgSO_4$) for 30 min in humidified atmosphere with 5% $CO_2$ at 37 °C. Thereafter, inserts containing the islets were moved to a fresh low-glucose (1.67 mM) KRBB, for 1 h and then transferred to a high-glucose (16.7 mM) KRBB and incubated for 1 h. Finally, the supernatant was collected and the insulin content was measured by using a mouse insulin ELISA (Mercodia Uppsala, Sweden) and a mouse insulin AlphaLISA (Perkin Elmer, Waltham, MA, USA) and was expressed as the stimulated-to-basal ratio of insulin secretion (stimulation index, SI).

### 2.5. Immuno-Protection of Pancreatic Islets

One day after pancreatic islets isolation (see the Supplementary Materials for further details), they were collected from the culture dish and washed once in PBS. Then, the pancreatic islets were encapsulated in a PEGDA hydrogel, as described in Supplementary Materials. Acryl-PEG functionalized with Anti-GFP was linked to the coating through photopolymerization. Briefly, a solution of Acryl-PEG-Anti-GFP 1 nM and Darocur$^®$ 1173 (0.3% *w/v*) was irradiated for 3 min under UV light (365 nm at 2 mW cm$^{-2}$) on Islet-hydrogels, obtaining functionalized hydrogels. The presence of Acryl-PEG- Anti-GFP on islet-hydrogels was then detected through a secondary antibody (Alexa Fluor$^®$ 488 Anti-Chicken, Thermo Fisher Scientific), incubated for 1 h, at RT (1:250). Then, the samples were washed three times in PBS for at least 5 min prior to be imaged with a LSM 700 laser scanning confocal microscope (Carl Zeiss, Bucarest, Romania) equipped with a Transmission-Photomultiplier (T-PMT), 2 reflection/fluorescence (R/FL) detection channels and 3 excitation lasers (405, 488, and 555 nm).

### 2.6. Statistical Analysis

Data were expressed as means ± standard deviation (SD). Significant differences between sets of data were analyzed for statistical significance using, where appropriate, either one- or two-way analysis of variance (ANOVA) with a post-hoc Bonferroni test or unpaired t-test. A probability value of 95% ($p < 0.05$) was used as the criterion for significance.

## 3. Results and Discussion

### 3.1. Chemical Activation of the Substrate

The scheme of Figure 3 shows the functionalization steps of the gold surface to enable the oriented binding of the Fc region of the antibody to the PEG molecules.

First, the gold surfaces (1 cm$^2$) were derivatized with MUA at different incubation times and imaged by NC-AFM. The derivatization of gold for 12 h induces the formation of domains due to the partial coverage of the substrate with the thiol group of the organic acid (Figure 4b) as also shown by phase images (Figure 4e). Indeed, being sensitive to the chemical composition of the surface groups, it confirmed the formation of MUA domains on the gold surface [32]. Moreover, the AFM analysis shows the presence of occasional peaks of few nanometers attributable to the formation of molecular aggregates, likely induced by the interactions between the alkyl chains ending to thiol aggregates [33].

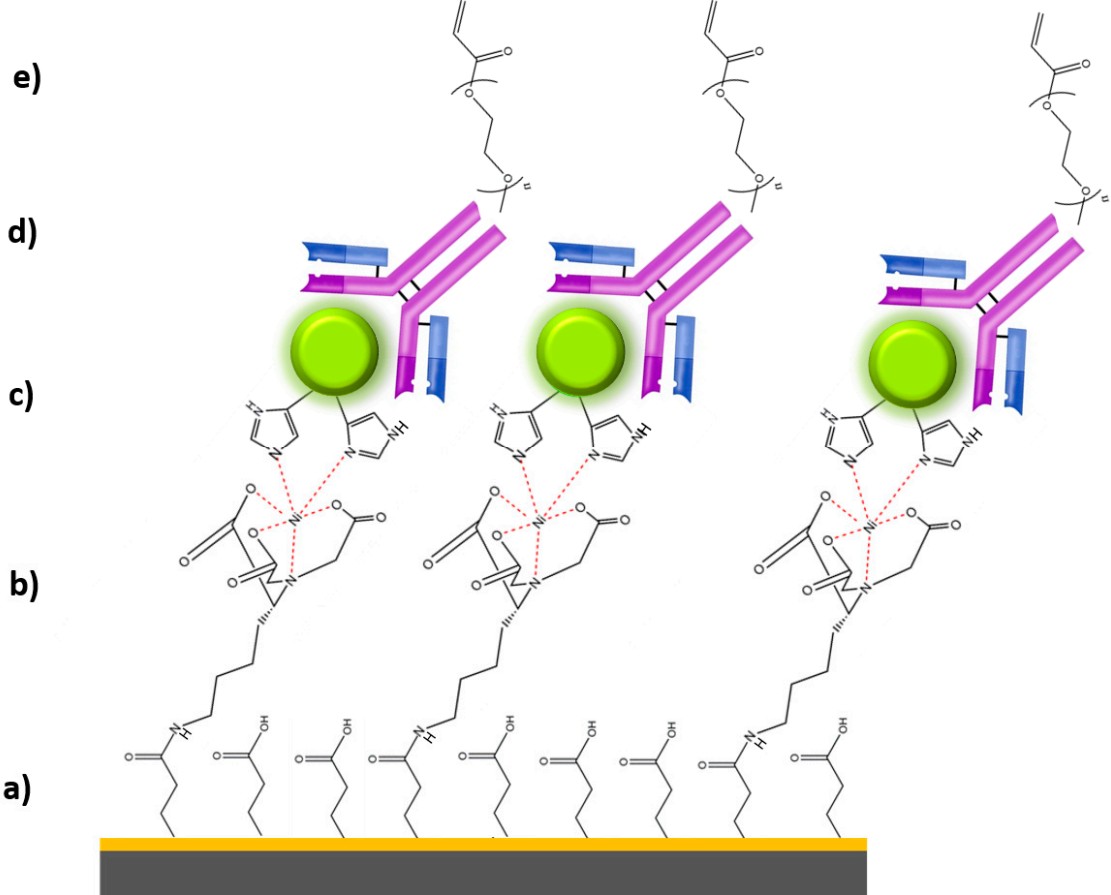

**Figure 3.** The scheme shows the steps of the site-directed functionalization of the antibody. (**a**) Coating of the gold surface with the organic acid MUA; (**b**) binding of NTA-lysine and complexation with $Ni^{2+}$; (**c**) functionalization with the His-tagged antigen. (**d**) The immobilized protein was recognized by the target antibody. (**e**) Finally, the constant fraction of the antibody was functionalized with Acryl-PEG-NHS.

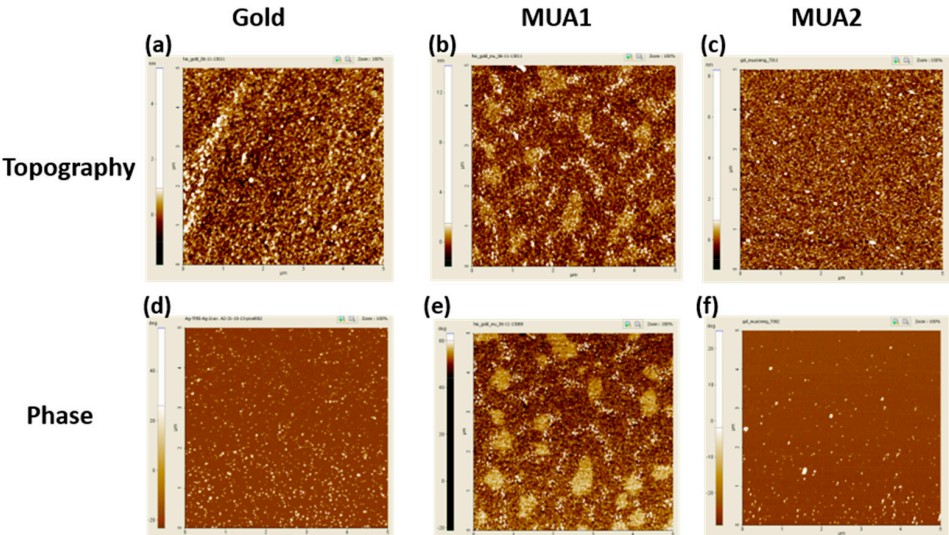

**Figure 4.** Atomic Force Microscope (AFM) topographic (**a–c**) and phase (**d–f**) images of gold surface on silicon chips (**a,d**), incubated over night with MUA (MUA1) (**b,e**) and coated with gold and incubated over weekend with MUA (MUA2) (**c,f**). The scan area is $5 \times 5$ µm.

The overall surface aspect varied significantly with a longer incubation (60 h), resulting in a homogeneous layer with a mean roughness value (Rq) of 0.7 nm (Figure 4c) and this result was established by the phase image (Figure 4f) that showed a complete coverage of the surface.

Contact angle measurements were performed to further investigate the MUA coating on gold surface (Figure S1). The contact values on the modified gold substrates significantly decreased compared to that measured for the bare gold ($p < 0.01$). In particular, thiol derivatization reduced contact angle from 98° of gold surface to 69° after 60 h, thanks to a better homogeneity of functionalization, as already verified by AFM analysis. NTA surface modification further reduced the contact angle of the water droplets from 69° to 63°, increasing wettability of the SAM.

As shown in the Scheme, NTA-lysine was then bound to the carboxyl-ending group of MUA followed by nickel ions complexation.

## 3.2. GFP Immobilization

The NTA-lysine-Ni$^{2+}$-modified surfaces have been widely used to target hexahistidine (His6) proteins [34]. Indeed, oligo-histidine (His) sequences in combination with NTA-lysine-Ni$^{2+}$ are employed for purification through chromatographic techniques since the binding is highly selective as well as reversible [35,36]. In this study, we exploited this coordination chemistry to bind the histidine residues of the His-tagged GFP to NTA-lysine-Ni$^{2+}$ modified surfaces in a reversible manner.

To estimate the amount of protein attached to the substrate, it was exposed to imidazole buffer. Indeed, the addition of a competitive ligand, such as histidine or imidazole, or a chelating agent, such as EDTA, enables the triggered release of His-tagged ligand from the surface [37]. Remarkably, such a reversible functionalization results in the reusability of the functionalized supports for successive experiments. The detached GFP, named "eluted GFP", was quantified by both fluorescence analysis and Western Blot.

In the case of the fluorescence analysis, the curve of the eluted protein was compared to the recovered fraction (that corresponds to the unbound protein amount) and to the feeding solution of GFP. The plot of the fluorescent intensities is reported in Figure 5a.

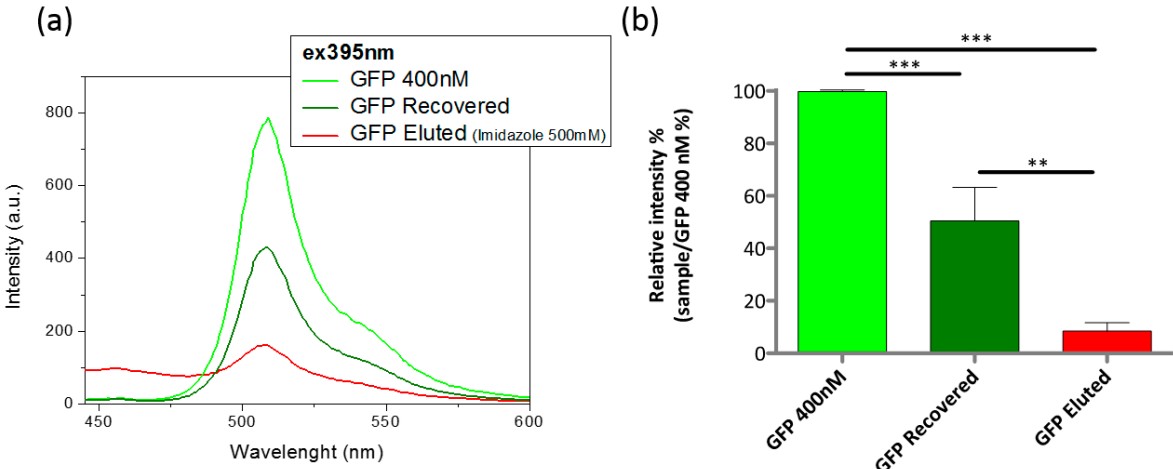

**Figure 5.** (**a**) Fluorescence emission spectra and relative intensity percentage of green fluorescent protein (GFP) solutions of solutions (excitation 395 nm, emission 508 nm) as quantitative estimation of the protein recovered after incubation (dark green curve) and eluted from the solid support (red curve) compared to the feeding solution of GFP (light green curve) (**b**). The quantitative values have been averaged over 3 measurements. Error bars represent standard deviation (SD) (*** $p < 0.001$, ** $p < 0.01$).

As shown in Panel (b) of Figure 5 the decrease in fluorescence intensity of recovered GFP (dark green bar) when compared to the feeding solution (light green bar) was statistically significant ($p < 0.001$), thus indicating that about 50% of the protein was immobilized onto the surface [38].

However, the fluorescence intensity at 508 nm of the eluted GFP (red bar) had a relative intensity of about 8.4%. To explain this discrepancy, it is plausible that not all the protein was detached and eluted, and it is likely that a fraction of the eluted GFP could be quenched. Fluorescence quenching may occur when the fluorophore is partially denatured or because it is unprotected from quenching by jostling water dipoles, paramagnetic oxygen molecules, or cis-trans isomerization [39,40]. Moreover, it is reported that wild-type GFP is influenced by pH variations; indeed, it is quenched by acidic pH values, whereas at basic pH, it is absorbance and excitation amplitude decreases [41–43].

To confirm these findings, the fluorescence estimation of GFP was compared to the densitometry analysis of the Western Blots (WB) bands [44]. Figure 6a shows the WB analysis of the three fractions, whereas in Figure 6b the densitometric values are reported as percentages of the control values.

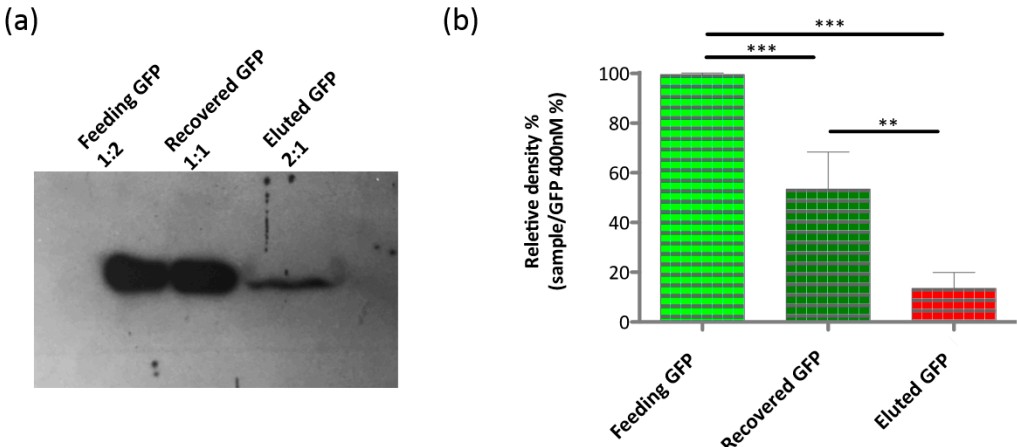

**Figure 6.** Western blot analysis (**a**) and relative density percentage (**b**) performed on the feeding solution of GFP (light green bar), GFP recovered after incubation (dark green bar), and GFP eluted from the solid support (red bar). The reported values have been averaged over 3 measurements. Error bars represent standard deviation (SD) (*** $p < 0.001$, ** $p < 0.01$).

Quantification of the eluted GFP by WB indicates that about 13.5% of initial GFP protein was detached from the substrate. These results are comparable to those from the fluorescence analysis, and confirm that imidazole 500 mM seemed to be not able to elute all the immobilized GFP and that GFP could be partially quenched [41]. The estimation of the protein density per surface unit was equal to 54 nmol cm$^{-2}$.

*3.3. The Acryl-PEG-Anti-GFP Complex*

The next functionalization step consists in the binding of the antibody to the immobilized antigen followed by the conjugation of Acryl-PEG to the Fc fragment of the antibody via EDC chemistry. Once the directional binding of Acryl-PEG to the antibody was complete, the PEG-Anti-GFP complex had to be detached from the substrate to proceed toward the development of the PEG hydrogel by breaking the Ag-Ab pair. The specific binding between antigens and antibodies involves weak interactions, i.e., ionic bonding, hydrogen bonding, and van der Waals attractions. The strength of Ag-Ab complexes depends on the relative affinities and avidities of the antibodies [29]. To dissociate the Ag-Ab binding and to recover the antibody at a high yield, purity, and stability, an elution buffer containing 0.1 M glycine-HCl pH 2.8 was used. Afterwards, the complex PEG-Anti-GFP was quantified by WB. Panel (a) of Figure 7 reports the WB analysis, while panel (b) shows the relative density percentage of the eluted Acryl-PEG-Anti-GFP complex compared to that of the recovered Anti-GFP and the initial amount of Anti-GFP 200 nM added to the substrate.

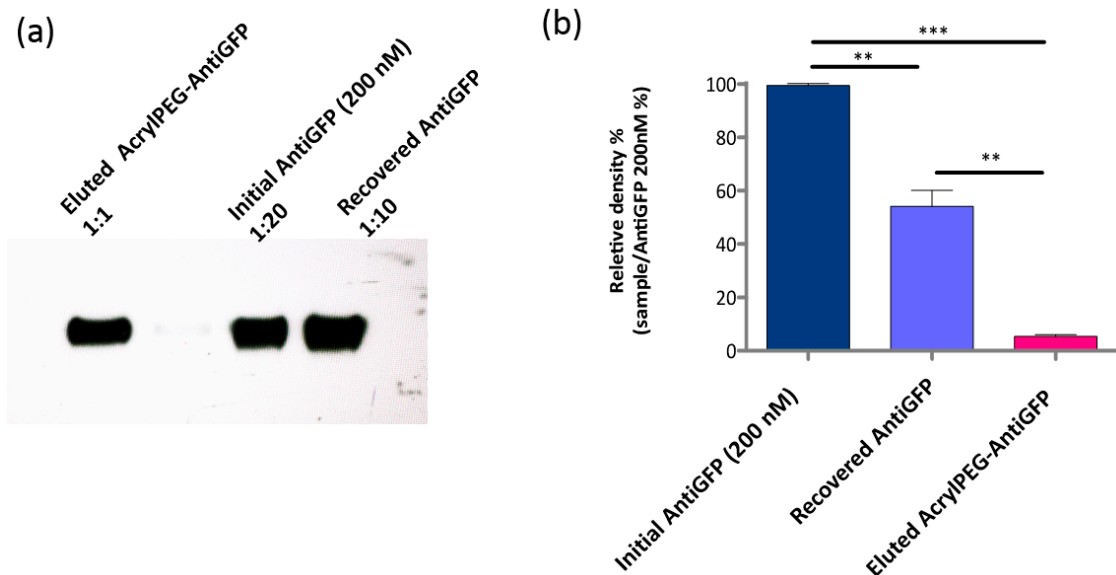

**Figure 7.** Western blot analysis (**a**) and relative density percentage (**b**) of mother solution of Anti-GFP (blue bar), Anti-GFP recovered after incubation (light blue bar), and PEG-Anti-GFP eluted from the solid substrate (pink bar). Reported values have been averaged over three measurements. Error bars represent standard deviation (SD) (*** $p < 0.001$, ** $p < 0.01$).

WB analysis suggests that about 46% of the initial amount of Anti-GFP is coupled to GFP immobilized on the functional substrates, whereas only a small percentage (8.3%) was collected after the elution process.

Unlike the elution of GFP with imidazole that did not require any post treatment, the detachment of Acryl-PEG-Anti-GFP using 0.1 M glycine-HCl pH 2.8 required a fast neutralization of the pH, to avoid acid-induced denaturation [29]. After the pH neutralization, Acryl-PEG-Anti-GPF was washed in PBS and concentrated by ultrafiltration with a 50 kDa centrifugal filter device. This greater number of steps might explain the low efficiency of elution process together with the fact that 0.1 M glycine-HCl buffer pH 2.8 only partially dissociates antigen-antibody complexes [45].

It is also worth noting that the mass contribution of Acryl-PEG to the eluted PEG-Anti-GFP complex cannot not be appreciated due to the low MW of the polymer (around ≈3 kDa) as compared to the whole complex (Mw ≈ 180 kDa) (Figure 7a).

### 3.4. Acryl-PEG-Anti-GFP Photo-Crosslinking on PEG Hydrogel Using a Selective Mask

To assess the ability of the acrylic group of the Acryl-PEG-Anti-GFP complex to bind the hydrogel membranes, an immunostaining study with a FITC-labeled secondary antibody was performed after a selective conjugation of the complex to the hydrogel. To this aim, a TEM grid was used as UV photomask to limit the areas of photopolymerization and to exclude a non-specific adsorption of the antibody on the dish.

As shown in Figure 8, the Acryl-PEG-Anti-GFP eluted from the solid support was selectively conjugated to the PEGDA hydrogel during photo-crosslinking since it is detectable only in the regions exposed to UV light.

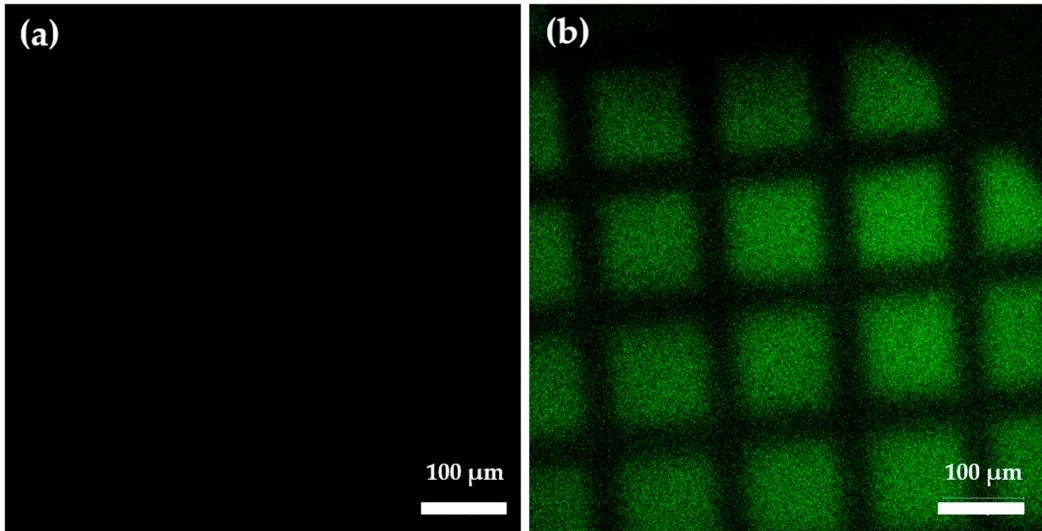

**Figure 8.** Confocal microscopy images of PEGDA hydrogel (**a**) and PEGDA hydrogel functionalized with Acryl-PEG-Anti-GFP (**b**) stained with Alexa Fluor® 488 anti-chicken. Scale bars are 100 μm.

### 3.5. Immuno-Encapsulation of Pancreatic Islets

The presence of the Acryl-PEG-Anti-GFP coating on the surface of the pancreatic islet-hydrogels was visualized by confocal microscopy that confirmed the uniform coverage of the pancreatic islets. The interfacial photo-crosslinking protocol was designed to obtain a conformal PEG-based coating around each islet (Figure S2). In Figure 9, pancreatic islet-hydrogels were stained with a secondary Anti-Mouse IgG (Fab specific)-Alexa Fluor® 488-conjugated antibody. The green fluorescence (upper panels of Figure 9) associated to the Acryl-PEG-Anti-GFP evidences the homogeneous distribution of the coating around the islet. On the other hand, control (CTRL) islet that was not functionalized with the hydrogel, did not show any fluorescence signal (lower panels of Figure 9).

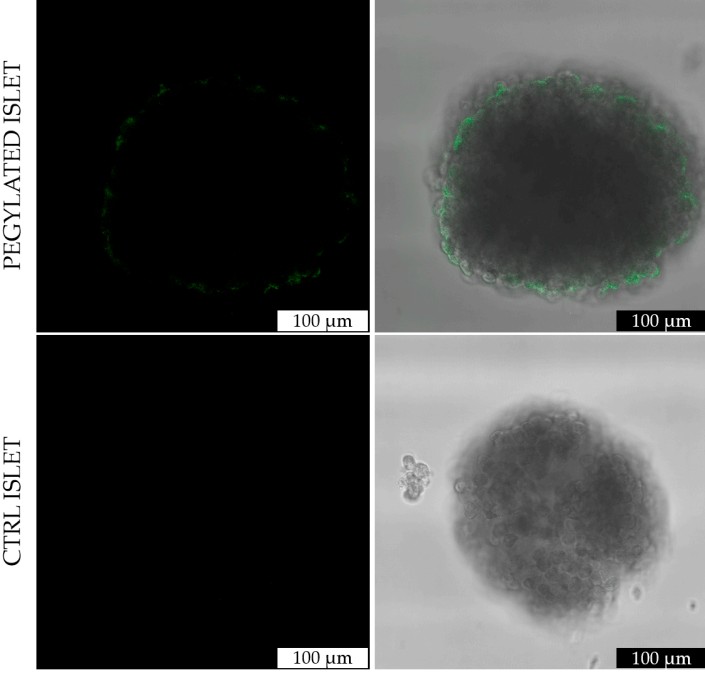

**Figure 9.** Confocal microscopy images showing (upper panels) an islet-hydrogel functionalized with Acryl-PEG-Anti-GFP and (lower panels) a CTRL islet stained with a secondary antibody (Alexa Fluor® 488 goat Anti-Chicken). Scale bar = 100 μm.

The encapsulation of pancreatic islets within the immuno-functionalized PEG hydrogel is expected to enhance islet cell survival and functionality and does not affect the islet viability as compared to control islets, as shown in Figure 10. Indeed, the viability qualitatively determined with FDA-PI staining 24 h post PEG functionalization did not show evident differences between control and functionalized islets.

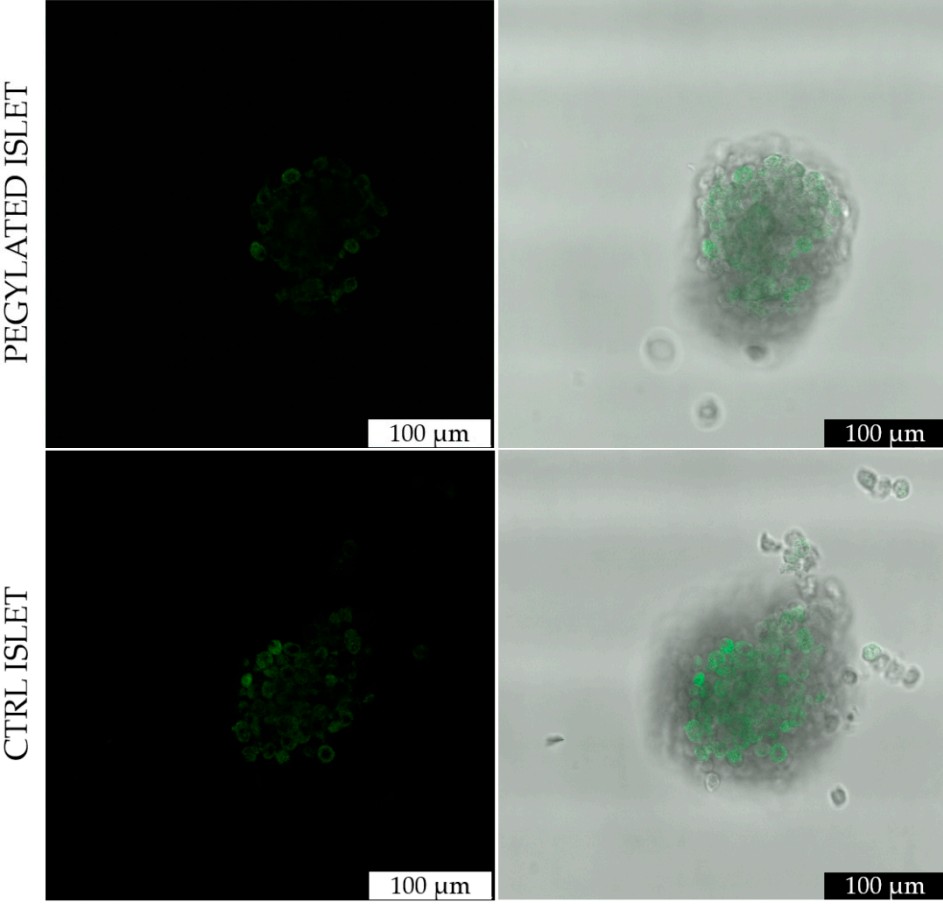

**Figure 10.** Confocal microscopy images of viable PEGYLATED (upper panels) and CTRL (lower panels) islets. The viability was qualitatively detected with the fluorescein diacetate (FDA) *versus* propidium iodide (PI) staining method 24 h post functionalization. Scale bar = 100 µm.

Moreover, the cellular functionality of the islets, in terms of response to glucose, was measured via a glucose-stimulated insulin secretion (GSIS) assay in vitro 24 h post-encapsulation. For all experiments, the insulin stimulation from basal to stimulated levels illustrated the expected response, with stimulated levels higher than basal ones. As shown in Figure 11, the insulin stimulation index was about 4, both for control and encapsulated islet, which, therefore, did not show significantly different behaviors. Therefore, the insulin secretion capacity of encapsulated islets in vitro was maintained compared to control islets.

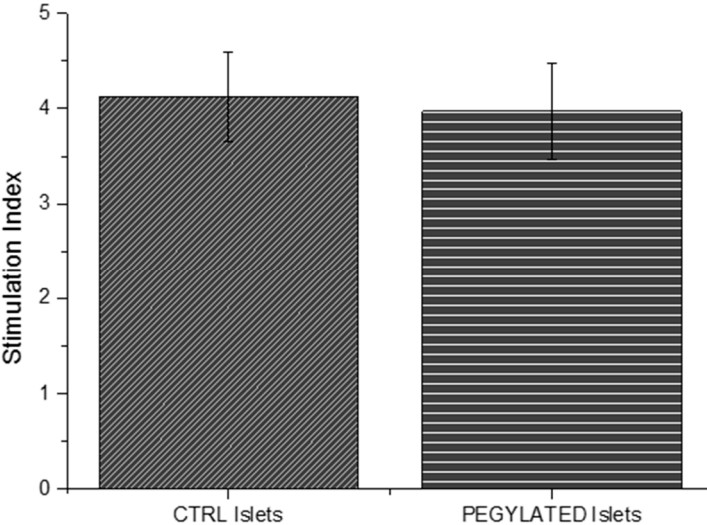

**Figure 11.** Results of glucose-stimulated insulin secretion (GSIS) assay on encapsulated islets compared to control islets on day 1 after encapsulation, for which the glucose stimulation index (SI) is the ratio of stimulated-to-basal insulin secretion (n ≥ 5).

### 3.6. Site-Directed Functionalization of an Immunorelevant Antibody

CTLA-4 was selected as immunorelevant antigen, as it is a major negative regulator of T cell responses [46]. In autoimmune diseases and graft rejection, CTLA-4 supports a dynamic but complex process of immune regulation that is prominent in the control of self-reactivity [47]. In a pioneering paper, it was demonstrated that CTLA4-Ig therapy was able to inhibit the rejection of human pancreatic islet in transplanted mice [48]. Following this work, other studies supported the use of CTLA4-Ig as an effective immunomodulatory to prolong allografts survival [26].

In this sense, an engineered fusion antibody (CTLA-4 Ig) that displays the extracellular domain of CTLA-4 and the Fc portion of a IgG, and that binds with high affinity the two ligands of CD28, respectively B7-1 (CD80) and B7-2 (CD86), was selected [49]. One of its ligands (B7-1 His-tagged) was employed for the immobilization protocol onto the substrate [50], while the CTLA-4-Ig was functionalized with the Acryl-PEG.

To evaluate the applicability of this functionalization strategy the Acryl-PEG-CTLA-4 Ig complex was detached using the same elution protocol (0.1 M glycine-HCl, pH 2.8, at 4 °C for 10 min); afterwards, WB was performed to detect quantitatively the amount of complex detached (Figure 12). As already observed in the case of the Acryl-PEG-Anti-GFP complex, the analysis did not reveal the complete elution of the Acryl-PEG-CTLA-4 Ig complex from the solid support, likely due to the strength of Ag-Ab binding between B7-1 and CTLA-4 Ig.

Therefore, the data presented so far show the potential use of this approach to develop a thin and biocompatible immunocoating for pancreatic islets. On the other hand, the methodology requires further optimization to improve the recovery yield of the Ab-functionalized coating and make it applicable to large-scale use.

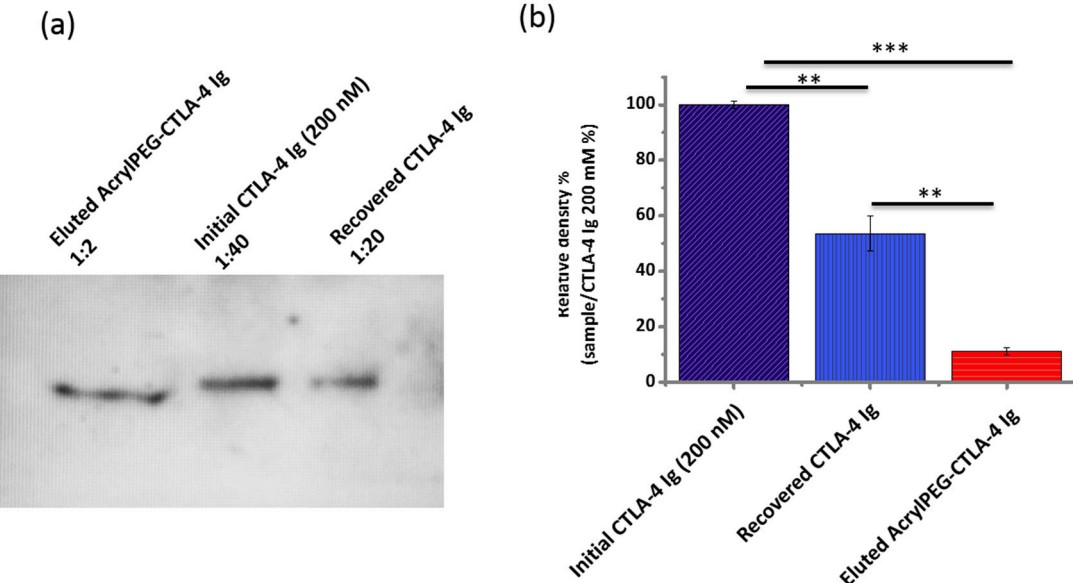

**Figure 12.** Western blot analysis (**a**) and relative density percentage (**b**) of mother solution of CTLA-4 Ig (dark blue bar), CTLA-4 Ig recovered after incubation (light blue bar) and Acryl-PEG-CTLA-4 Ig complex eluted from the solid substrate (red bar). Reported values have been averaged over three measurements. Error bars represent standard deviation (SD) (*** $p < 0.001$, ** $p < 0.01$).

## 4. Conclusions

In order to reduce the immunosuppression response in patients affected by diabetes 1 and transplanted with the pancreatic islets of a donor, several coating procedures of the donor islets have been proposed. Here we designed an immuno-functionalization strategy aimed to conjugate a therapeutically relevant antibody to PEG encapsulated islets. We proposed an approach based on the site-directed functionalization of the antibody while keeping functional the antigen-recognition site and thus preserving its functionality. Gold-coated silicon wafers were functionalized using 11-Mercaptoundecanoic acid and were used as a substrate for further stepwise modification, leading to a nickel(II)-terminated ligand surface. As a proof of principle, the specific immobilization was demonstrated for a generic fluorescent antigen antibody pair (GFPA–Anti-GFP), confirming the attachment of the His-tag protein to the solid support. The immobilized His-tagged protein was recognized by the target antibody, thus blocking the binding site; then the constant fraction of the antibody was functionalized with Acryl-PEG-NHS. The recovery efficiency of the PEG-antibody complex is not satisfying (less than 10% of the initial amount of antibody), thus requiring further implementation and modification of the protocol steps. Nevertheless, this approach proposes a controlled functionalization chemistry to preserve the binding site of the antibody.

Moreover, murine pancreatic islets were successfully encapsulated with the PEG hydrogel bearing the Acryl-PEG-Anti-GFP complex, and their viability and functionality was not affected by the PEG coating.

To show the general applicability of the approach, the immobilization protocol was extended was to a therapeutically relevant antigen-antibody couple, B7-1–CTLA4-Ig [25]. Even in this case, the recovery efficiency was quite low, suggesting that this methodology requires further optimization prior to become applicable in vivo.

In conclusion, this strategy represents an attractive ready-to-use tool for customizing a patient-specific immunotherapy, by adjusting the specificity of the conjugated immunotherapeutic agent, based on an autoimmune response. The application of this methodology to the immunocoating of pancreatic islets deserves research efforts to optimize the quantitative yield of the process.

**Supplementary Materials:** The following are available online at http://www.mdpi.com/2076-3417/10/17/6056/s1, Figure S1: Values of contact angle of either gold surface, gold surface after 12 h and 60 h incubation with MUA, or gold surface after 60 h incubation with MUA and NTA. Reported values have been averaged over 5 measurements. Error bars represent standard deviation (SD). (*** $p < 0.001$, ** $p < 0.01$). Figure S2: Schematic representation of the experimental design for the conformal encapsulation of islets through photopolymerization. Islets are incubated with the photoinitiator that adsorbs on the islet surface. The islet-photoinitiator complex is incubated with the PEGDA and exposed to UV to encapsulate the islet, inducing the crosslinking of the polymer.

**Author Contributions:** Conceptualization, L.B., A.Q., and M.M.; methodology, A.C., U.M., A.B., and T.V.; writing and editing, A.C., U.M., L.B., and A.Q.; supervision, A.S. All authors have read and agreed to the published version of the manuscript.

**Funding:** This research was partly funded by the Progetto regionale Lab on a Swab (cod. OTHZY54; funds to L.B.), and the University of Salento with the "Fondi del 5 per mille per la ricerca" (funds to M.M.).

**Acknowledgments:** The authors acknowledge Francesca Gatti for the technical support concerning the isolation of pancreatic islets.

**Conflicts of Interest:** The authors declare no conflict of interest.

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
