# Peer review of "Design of Antibody-Functionalized Polymeric Membranes for the Immunoisolation of Pancreatic Islets"

_applsci, doi:10.3390/app10176056_

Round 1

Reviewer 1 Report

The manuscript by Cavallo et al. aims at reporting a chemical method for controlled functionalization of antibodies to PEG for islet encapsulation that preserves the binding site of the antibody. The work is of potential interest for the field of beta cell replacement for treatment of patients with type-1 diabetes. However, in its current form, the report does not provide significant data to demonstrate applicability for encapsulation of pancreatic islets and for functionalization with immunomodulatory molecules. Impact of the work can be increased if additional data on islet encapsulation are provided, including extensive characterization of antibody functionalization on islet surface, islet functionality (as glucose-stimulated insulin secretion after coating), stability of functionalized coatings during long-term culture. Alternatively, the report could be focused on the methodology and provide feasibility functionalization with immunomodulatory antibodies. Of note, abatacept (CTLA4-Ig) is a fusion protein and not an antibody. The poor recovery efficiency of the PEG-antibody complex (<10%) is of concern and should be discussed if the report is refocused on the methodology. Additional comments below:

Abstract

  • Risk of ‘rejection’
  • Define Ab
  • Include results in addition to methods in the abstract

Keywords:

  • Revise Polyethylene glycol

Introduction:

  • 2 line 45 ‘islet cells’
  • 2. line 58 add ‘local’
  • Specify how Ag-Ab conjugation could be exploited for local immunomodulation

Figures:

  • Figure S3 show only viable (green) cells. It is unlikely that no dead cells (red) were found in any of the conditions analyzed.

Author Response

We are realy grateful to the reviewer for the accurate reading of the manuscript.

The raised points have been addressed in the revised version of the manuscript.

Reviewer 2 Report

The authors reported a strategy for encapsulating islets within a PEG hydrogel coating functionalized with an immunosuppressive antibody to create both passive and active barrier to the host immune system. However, no actual immunosuppressive antibody was conducted in this work and the immunosuppressive function of this system is unknown. Some more questions should be addressed as below.

  1. The manuscript title is overstated. First, this approach is not novel. Second, the method was just conducted using a nonfunctional GPF protein model, and no actual immunosuppressive antibody was evaluated. And the same to the conclusion, inflated the significance.

  1. The chemical structures in Figure 1 are not clear, a larger text size should be used. And the hydrogen H was missing in amido bond (-CO-NH-) in figure 1b and 1c. Some of the writing of chemical names are not accurate. For example, the nitrogen element (N) should be in italic (N). The first key word “polyethylen glycol” has a typo, the correct one is polyethylene glycol.

  1. The recovery efficiencies of GFP and AcrylPEG-AntiGFP complex are quite low. More importantly, based on the presented data, the low recovery efficiency of GFP is more likely from the protein denaturation instead of the fluorescence quenching. As mentioned earlier, the immunosuppressive antibody activity remains uncertain after all these procedures.

  1. A crosslinked PEGDA hydrogel shell was applied on islet surface before the conjugation of the AcrylPEG-AntiGFP complex. But the PEGDA hydrogel shell is barely detectable in both Figure 9 and Figure S3. The characterization of the PEGDA hydrogel coating need be provided.

  1. The viability test of the islets in Figure S3 is not robust. Generally, the green fluorescent of fluorescein diacetate in live cells are pretty strong, but quite dim green signal was showed in Figure s3 and no any red signal was detected from the propidium iodide. Moreover, the islets do not look healthy, the islet surface is rough and the cells in islets are loose. Glucose-stimulated insulin secretion (GSIS) and immunohistochemistry staining of insulin are better characterizations to evaluate the viability and function of islets.

Author Response

We thank the referee for the analysis of the manuscrit. In the revised version of the paper we have introduced new experimental data to address the requested revisions

Round 2

Reviewer 1 Report

The authors did not revise the manuscript as suggested. Minor concerns were not addressed and major concerns still remain. Applicability of the functionalization protocol with CTLA4-Ig immunomodulatory molecules is described but no data are provided. Also, the poor recovery efficiency of the PEG-antibody complex (<10%) is of concern and should be discussed if the report. For applicability to islet transplantation, it is critical to show that functionalization does not compromise islet viability and functionality. Thus, data showing that viability and glucose-stimulated insulin secretion after functionalization should be included in the manuscript.

Minor concerns still remaining and not addressed in the revised version:

The abstract has not been revised as suggested in the initial review:

  • Risk of ‘rejection’
  • Define Ab
  • Include results in addition to methods in the abstract
  • Update the abstract with new results provided

The keywords haven’t been revised as suggested in the initial review:

  • Revise Polyethylene glycol

The introduction has not been revised as suggested in the initial review:

  • 2 line 45 ‘islet cells’
  • 2. line 58 add ‘local’
  • Specify how Ag-Ab conjugation could be exploited for local immunomodulation. Of note, for immunosuppression CTLA4-Ig and not anti-CTLA4 is used. Antibodies against CTLA4 promote immunity and are used in cancer immunotherapy.

Author Response

Dear Editor and Dear Referees,

We apologize for the inconvenience: it was just a problem while managing the revised versions of the manuscript. By an error and due to technical issues, the version uploaded on the review platform was not the proper one. The correct file has been now uploaded. It contains the requested modifications: we have also tried to satisfy the points raised in the second revision and to make the results more comprehensive and clearer. In addition, we moved some of the data presented in the SI to the main manuscript, as requested.

Reviewer 2 Report

The authors did not address all comments. The revised manuscript does not include all changes as mentioned in the Cover Letter and some data are missing.

  1. The word “novel” was not removed from the manuscript title.
  2. No changes were made in Figure 1 to optimize the chemical structures. And some typos and inaccurate writing of chemical names were not corrected.
  3. More importantly, no related data were presented for the new experiments about the immune relevant Antigen-antibody pair, B71-CTLA-4.

Author Response

(The authors gave the same response as above.)

Round 3

Reviewer 1 Report

The authors have revised the manuscript as suggested and it is now suitable for publication.

Author Response

We thank the referee for the positive feedback: her/his contribution helped to improve the robustness of the experimental results and of the scientific discussion. 

Reviewer 2 Report

The authors have addressed all comments. The manuscript can be accepted after minor revision.

  1. Line 400: “WB was performed to detect quantitatively the amount of complex detached (Figure 10).” The corresponding figure is Figure 12 not Figure 10.
  2. Correction is necessary through the manuscript.

Subscript should be applied for some chemical names, such as CaCl2, KH2PO4, CO2, etc.

A space is need between the number and unit, such as 37 °C

Author Response

We renew the thanks to the Referee for the accurate analysis of the manuscript and for the modifications/corrections suggested.

In the current version of the manuscript, the typo errors have been corrected and the work has been read carefully to check throughout.